# Nasendoscopy to Predict Difficult Videolaryngoscopy: A Multivariable Model Development Study

**DOI:** 10.3390/jcm12103433

**Published:** 2023-05-12

**Authors:** Phillip Brenya Sasu, Jennifer-Isabel Pansa, Rupert Stadlhofer, Viktor Alexander Wünsch, Karolina Loock, Eva Katharina Buscher, André Dankert, Ann-Kathrin Ozga, Christian Zöllner, Martin Petzoldt

**Affiliations:** 1Department of Anesthesiology, Center for Anesthesiology and Intensive Care Medicine, University Medical Center Hamburg-Eppendorf, Martinistrasse 52, 20246 Hamburg, Germany; p.sasu@uke.de (P.B.S.); isabel.pansa@icloud.com (J.-I.P.); v.wuensch@uke.de (V.A.W.); k.loock@uke.de (K.L.); kohseeva@gmail.com (E.K.B.); a.dankert@uke.de (A.D.); c.zoellner@uke.de (C.Z.); 2Department of Otorhinolaryngology, University Medical Center Hamburg-Eppendorf, Martinistrasse 52, 20246 Hamburg, Germany; r.stadlhofer@uke.de; 3Institute of Medical Biometry and Epidemiology, University Medical Center Hamburg-Eppendorf, Martinistrasse 52, 20246 Hamburg, Germany; a.ozga@uke.de

**Keywords:** airway management, intratracheal intubation, laryngoscopy, laryngoscopes, videolaryngoscopy, laryngeal diseases

## Abstract

Background: Transnasal videoendoscopy (TVE) is the standard of care when staging pharyngolaryngeal lesions. This prospective study determined if preoperative TVE improves the prediction of difficult videolaryngoscopic intubation in adults with expected difficult airway management in addition to the Simplified Airway Risk Index (SARI). Methods: 374 anesthetics were included (252 with preoperative TVE). The primary outcome was a difficult airway alert issued by the anesthetist after Macintosh videolaryngoscopy. SARI, clinical factors (dysphagia, dysphonia, cough, stridor, sex, age and height) and TVE findings were used to fit three multivariable mixed logistic regression models; least absolute shrinkage and selection operator (LASSO) regression was used to select co-variables. Results: SARI predicted the primary outcome (odds ratio [OR] 1.33; 95% confidence interval [CI] 1.13–1.58). The Akaike information criterion for SARI (327.1) improved when TVE parameters were added (311.0). The Likelihood ratio test for SARI plus TVE parameters was better than for SARI plus clinical factors (*p* < 0.001). Vestibular fold lesions (OR 1.82; 95% CI 0.40–8.29), epiglottic lesions (OR 3.37; 0.73–15.54), pharyngeal secretion retention (OR 3.01; 1.05–8.63), restricted view on rima glottidis <50% (OR 2.13; 0.51–8.89) and ≥50% (OR 2.52; 0.44–14.56) were concerning. Conclusion: TVE improved prediction of difficult videolaryngoscopy in addition to traditional bedside airway examinations.

## 1. Introduction

Although airway management problems are a main reason for anesthesia-related adverse events and liability claims against anesthetists [1,2,3,4], established bedside airway examination tests only show poor predictive performance [5,6]. Patients with pharyngolaryngeal lesions such as tumors, abscesses, edema or hyperplasia are at high risk for difficult airway management [7,8,9,10,11] but most frequently require general anesthesia [7,8,9,11,12]. Nevertheless, most bedside airway examination tests, such as the upper lip bite test, Wilson score or Simplified Airway Risk Index (SARI) solely rely on anatomic and functional assessments of the head–neck region and jaw joint and ignore pharyngolaryngeal lesions [5,6,13,14]. Hence, pharyngolaryngeal lesions represent a faithful blind spot in these traditional bedside airway examination scores [9].

Transnasal videoendoscopy (TVE) is standard of care for the detection, mapping and staging of pharyngolaryngeal lesions [15,16]. Beyond this, TVE improves prediction of difficult intubation with direct laryngoscopy [9,17,18]; however, many studies excluded patients with pharyngolaryngeal lesions [17,18]. In terms of invasiveness, time and costs, TVE cannot be used for screening. A rational concept for preselecting individuals that might benefit from additional preoperative TVE is yet to be determined, and it is still unclear how TVE findings can be used in addition to traditional bedside airway examination tests.

Despite limited data, current guidelines recommend TVE particularly in individuals with known or suspected obstructing glottic or supraglottic airway pathologies [19,20]. As many patients with pharyngolaryngeal lesions have typical clinical signs such as dysphonia, dysphagia, cough or stridor [15,16], a question arises: can these clinical factors be used to preselect high-risk patients, or even as a substitute for preoperative TVE?

Although videolaryngoscopy has revolutionized airway management [19,20,21,22,23], most established bedside airway examination tests have only been developed for direct laryngoscopy [5,13,14]. Still very little is known about how to predict difficult videolaryngoscopic intubation [24,25,26,27]. Previous difficult airway management is the most accurate predictor of future difficulty [28,29]. A universal classification for videolaryngoscopy—the VIDIAC score—has been introduced recently and is a validated reliable tool for reproducible, scalable recordings of videolaryngoscopic findings [11,30]. There is growing evidence that videolaryngoscopy is useful to avoid failed intubation, hypoxemic events and accidental esophageal intubation while improving glottic view [23]. Videolaryngoscopy became more universally available in many hospitals and regions [31,32], and it has been recommended using videolaryngoscopy routinely whenever possible [33].

It is unknown if TVE is able to predict difficult videolaryngoscopic intubation, and the question remains if there is a benefit of preoperative TVE in terms of efficiency, costs and healthcare resources if easy-to use videolaryngoscopes are universally available at the bedside.

This multivariable model development study aims to determine if TVE examinations or symptom screening or both improve the performance of the SARI when predicting difficult videolaryngoscopic intubation in patients with anticipated difficult airway management undergoing ear, nose and throat (ENT) or oral and maxillofacial (OMF) surgery.

## 2. Materials and Methods

The Videolaryngoscopic Intubation and Difficult Airway Classification (VIDIAC) trial is a single-center prospective model development study, performed in accordance with the Declaration of Helsinki. The study design, conduction, and reporting were carried out in accordance with the STROBE statements [34]. The study was approved by the Ethics Committee of the Medical Association of Hamburg (PV5856, 10 August 2018, amendment 12 August 2019), and registered with ClinicalTrials.gov (NCT03950934). Participants gave written informed consent. The present findings result from an analysis of an independent dataset within the VIDIAC study [11], as outlined in the study protocol.

### 2.1. Patient Allocation and Data Collection

Adult patients presenting at the University Medical Center Hamburg, Eppendorf, Anesthesiology Preassessment Clinic before elective ENT or OMF surgery between 1 April 2019 and 3 April 2020 were assessed for eligibility (Figure 1). Patients received a structured preoperative airway risk assessment in line with the standards laid out by the Department of Anesthesiology, which included (but was not limited to) physical examination, SARI [13], medical history, upper lip bite test [5,6], specific clinical factors (such as dysphagia, dysphonia, cough, stridor, age, sex and height), history of head and neck radiotherapy, pharyngolaryngeal lesions and TVE, if feasible.

Preoperative TVE examinations were performed by skilled ENT physicians or a skilled senior consultant anesthetist (MP); videos were captured (Viewpoint 5, GE Company, Boston, MA, USA) and systematically reviewed in a blinded fashion based on predefined criteria, as previously reported [9]. Only patients with TVE examinations no older than 90 days before surgery without progression of the underlying disease or clinical symptoms within this time frame were included in the TVE sub-cohort.

Study assessments and outcome variables were recorded separately from clinical notes to allow multiple independent assessments for participants who received multiple anesthetics.

### 2.2. Eligibility Criteria

Adult patients with expected difficult airway management with tracheal intubation aided by videolaryngoscopy were included. Patients with planned awake tracheal intubation or pregnant women were excluded. Following our in-house standards, videolaryngoscopes with Macintosh-type blades (C-MAC^TM^, Karl Storz, Tuttlingen, Germany) were used first line in all participants. Anesthesia induction, patient positioning, airway optimization maneuvers, tracheal intubation, use of airway adjuncts, and conversion to different intubation techniques and devices—for instance, direct epiglottic lifting or transition to hyperangulated blades or flexible bronchoscopes—were left at the discretion of the anesthetist.

### 2.3. Sample Size Analysis

The method of Riley et al. [35] was used to calculate the sample size for model development studies (R package “pmsampsize”, version 1.1.0, R Foundation for Statistical Computing, Vienna, Austria). Assuming a difficult videolaryngoscopic tracheal intubation rate of 45% (determined via planned interim analysis after 100 observations, IRB amendment 12 August 2019), a shrinkage of predictor effects of 10% and a small optimism in apparent model fit, 400 anesthetics were included in the VIDIAC study to reach a sample size of 381, assuming 5% dropouts. A Cox–Snell R^2^ of 0.5 and 16 candidate predictors were assumed to be appropriate [35].

### 2.4. Outcome Measures

The primary outcome was that the anesthetist expected future videolaryngoscopic tracheal intubations to be difficult, which was documented as a difficult airway alert.

Secondary outcomes: transition to hyperangulated blade; transition to bronchoscopic intubation; difficult videolaryngoscopy [19]; difficult intubation [19]; numbers of laryngoscopy and intubation attempts; first pass success (only one attempt at laryngoscopy and intubation); time to tracheal intubation; first end-tidal carbon dioxide partial pressure after intubation; airway-related adverse events; length of hospital stay and in-hospital mortality.

Airway-related adverse events were defined as laryngospasm, bronchospasm, airway or oral trauma, including bleeding and dental injury, glottic swelling or use of corticosteroids to reduce swelling risk, esophageal intubation, oxygen saturations < 93% or unanticipated ICU admission [3].

### 2.5. Co-Variables for Model Fitting

In the first step, potentially eligible co-variables (predictor variables for model development) were identified via literature review [7,8,12,36,37,38,39], previous studies [9,40] and clinical considerations and comprise three categories:(i)Simplified Airway Risk Index [13]: the SARI score (0 to 12 points) encompasses seven binary or categorized variables: mouth opening, thyromental distance, Mallampati score (modification by Samsoon and Young [41]), neck mobility, mandibular protrusion, body weight and history of difficult intubation.(ii)Clinical factors (symptom screening): Typical clinical signs for pharyngolaryngeal lesion and demographic data were systematically assessed and subdivided into five sub-groups:
Dysphagia (self-reported; y/n): dysphagia; pharyngeal pressure or globus sensation; pharyngeal foreign body sensation; excessive salivation; odynophagia; frequent choking; difficulties swallowing liquids; food intake impossible.Dysphonia (self-reported and physical examination; y/n): altered voice; lumped speech; frequent throat clearing; weak voice or phonation difficulties; whispering or aphonia; progression of dysphonia in the last 3 months.Cough (self-reported and physical examination; y/n): dry cough; productive cough; impaired expectoration.Stridor (clinical examination with auscultation; y/n): inspiratory stridor.Demographic data: sex (male/female); age (years); height (cm).(iii)TVE findings: TVE examinations were systematically reviewed as previously reported [9]. Co-variables were subdivided into two sub-groups:
Location of lesions (y/n): hypopharynx; supraglottic; arytenoids; vocal cords; vestibular folds; epiglottis; base of the tongue; multiple unilateral findings; bilateral findings; no lesions.Accompanying findings (y/n): vulnerable mucosa with or without active bleeding; pharyngeal secretion retention; impaired vocal cord mobility; view restrictions on the rima glottidis due to lesions (none/relevant view restriction that cover <50%/≥50% of the glottis cross-sectional area).

Regarding our primary research question, SARI was used as a fixed variable in all three models without further preselection (forced variable). Exclusive least absolute shrinkage and selection operator (LASSO) regression analysis was used to select potentially eligible co-variables. Notably, penalized or regularized regression techniques such as LASSO regression can be used for variable selection to avoid model overfitting and optimism bias. LASSO regression identifies the variables and corresponding coefficients from a set of candidates. This leads to a model with minimized prediction error by imposing a constraint on the model parameters that shrinks the regression coefficients of more irrelevant variables towards zero. Variables with a regression coefficient of zero after shrinkage are excluded [42].

A ten-fold cross-validation was applied to check the robustness. Only coefficients that were not shrunk to zero were considered relevant. Thus, at least one variable was selected from each sub-group. For the selection of eligible clinical factors, the entire dataset was used (study cohort, *n* = 374) while the selection of eligible TVE parameters relied on the data of the TVE sub-cohort (*n* = 252). For more than one observation per patient, interventions were assumed to be independent of the anesthetist and patient. Thus, heterogeneity between anesthetists was not considered for variable selection.

### 2.6. Descriptive Statistics

Sample characteristics are given as absolute and relative frequencies, mean (standard deviation) and median (interquartile range, IQR), whichever was appropriate. Data were analyzed using SPSS 27 (IBM Inc., Armonk, NY, USA) and R 4.0.3 (R Foundation for Statistical Computing, Vienna, Austria).

### 2.7. Development of Three Multivariable Mixed Logistic Regression Models

Only variables that were selected via LASSO regression were used for fitting of the multivariable mixed logistic regression models. In the first step, a logistic regression analysis for the primary outcome measure was performed, with the SARI score being the only independent variable. Subsequently, the SARI was used as a component of all multivariable mixed logistic regression models, but also as a comparator.

To evaluate the incremental value of clinical factors and TVE in addition to the SARI, three different multivariable models were fitted: model A (SARI with clinical factors), model B (SARI with clinical factors and TVE) and model C (SARI with TVE). The dataset of the study cohort (initial model A and sensitivity analysis) and TVE sub-cohort (model A–C) was used for modeling. Missing data were not imputed (listwise deletion). A random effect was included to account for multiple anesthetics within one patient. Models were fitted by optimizing the restricted maximum likelihood criterion using an iterative nonlinear optimization algorithm [43]. Odds ratios (OR) for the fixed effects are presented and respective 95% confidence intervals (95% CI) were calculated via Wald approximation. As this is an explorative study, neither model validation nor adjustment for multiple testing were performed.

### 2.8. Sensitivity Analysis

To find an optimal trade-off between the goodness of model fit and simplicity of model A, we performed a sensitivity analysis using the data of the entire study cohort. Starting with the initial model A, which comprises all selected clinical factors together with the SARI, we critically appraised single clinical factors and determined if a reduction in redundancy between clinical factors or a removal of rare clinical factors would be accompanied with a relevant increase in goodness of model fit. We finally presented a simplified model A that was used for further analysis. To achieve this goal, we calculated the Akaike information criterion with correction for small sample size (AICc) before and after removal of single suspect variables and favored the model with the best goodness of fit.

Notably, the AICc is a statistical method used to determine the relative quality of a model; hence, AICc is often used to compare different models regarding their goodness of fit in contrast with their simplicity. To achieve this, AICc uses the maximum likelihood estimate and adds a penalty term for the number of independent variables. Thus, it handles the risks of overfitting and underfitting a model, resulting in a model with the greatest amount of variation using the fewest possible co-variables [44].

### 2.9. Comparison between Models

Model performance was compared between all models and the SARI using likelihood ratio tests (LRT). For this comparison, we report nominal *p*-values without correction for multiplicity. To estimate and compare the goodness of fit of the models in relation to model simplicity, the AICc was calculated and compared between the models and the SARI. Comparisons between models rely on the TVE sub-cohort dataset.

## 3. Results

It was determined that 374 anesthetics in 320 participants fulfilled all eligibility criteria and were subsequently analyzed (Figure 1). Suitable TVE examinations were available for 252 anesthetics (TVE sub-cohort). The baseline characteristics are given in Table 1.

In 183 of the 374 anesthetics (48.9%) the anesthetists issued difficult airway alerts after videolaryngoscopy (primary outcome measure) (Table 2). In five cases (1.3%), videolaryngoscopy was abandoned and flexible bronchoscopy was successfully used.

### 3.1. Development of Three Multivariable Mixed Logistic Regression Models

Exclusive LASSO regression with a ten-fold cross-validation was used in each category and identified 16 eligible co-variables (9 clinical factors and 7 TVE findings) that were used for modeling in addition to the SARI (Table 3). Within the TVE findings, vocal cord lesions, vestibular fold lesions, epiglottis lesions, multiple unilateral findings, pharyngeal secretion retention, glottis view restrictions <50% and ≥50% of the glottis area were selected.

Model A: The nine clinical factors were used for modeling of the initial model A. Sensitivity analysis was performed to simplify the model. Four clinical factors that did not relevantly improve the goodness of model fit were removed; hence, the final simplified model A comprises five clinical factors, together with the SARI. Based on the AICc, the final model is better than the initial one (Electronic Appendix A).

Model B encompasses the SARI, the five selected clinical factors and seven selected TVE variables. 

Model C encompasses the SARI and the seven preselected TVE variables. As we could not identify any redundancy between TVE variables, further sensitivity analysis was not performed.

### 3.2. Comparison between Models

Logistic regression analysis revealed an OR for the SARI of 1.33; 95% CI 1.13 to 1.58 (TVE sub-cohort) for primary outcome prediction (Table 4). Thus, with each additional point in the SARI score (0–12), the primary outcome probability increased by more than 30%. The ORs and 95% CIs of all co-variables in models A–C are given in Table 4 and Figure 2.

The LRT improved when clinical factors (*p* = 0.01) or TVE parameters (*p* < 0.001) were added to the SARI (Table 4). However, SARI combined with TVE parameters showed a better LRT than SARI combined with clinical factors (model A versus model C: *p* < 0.001). Moreover, adding clinical factors to model C (SARI with TVE) did not further improve the LRT (*p* = 0.37).

The AICc (lower values better) of the SARI (327.1) improved when clinical factors (model A: AICc 323.2) or clinical factors and TVE parameters were added (model B: AICc 316.8), but the best model fit was found for the SARI combined with TVE parameters (model C: AICc 311.0). Thus, model C was considered our final, optimal model.

## 4. Discussion

This study intended to determine if patients with expected difficult airways might benefit from additional preoperative TVE examinations or symptom screening for pharyngolaryngeal lesions. Is it possible to predict difficult videolaryngoscopy and to identify individuals that might benefit from awake tracheal intubation? Or can we be sure that we will be able to manage upcoming difficult intubation problems with universally available videolaryngoscopes at the bedside and thereby avoid additional costs and time?

Most traditional bedside airway examination tests including the SARI have been developed for direct laryngoscopy [5,6] and only few studies validated SARI for videolaryngoscopy [13,25,27]. 

We used AICc and LRT to identify the best fitting and performing model for the prediction of difficult airway alerts after videolaryngoscopy based on SARI, clinical factors and TVE findings. We found that the SARI also predicts difficult Macintosh videolaryngoscopy and that additional preoperative symptom screening or TVE further improved this prediction. Preoperative TVE provides more incremental diagnostic value than simple symptom screening. Our data indicate that TVE is a valuable diagnostic tool in suspicious patients with a high pretest probability for pharyngolaryngeal lesions. Thus, we believe that preoperative TVE should be performed whenever reasonable in individuals with anticipated difficult airways and suspected pharyngolaryngeal lesions, even if videolaryngoscopy is available at the bedside. Vestibular fold and epiglottis lesions, pharyngeal secretion retention and relevant view restrictions on the glottis were concerning TVE findings.

A previous study demonstrated that TVE has relevant implications for airway management planning [37]. Despite limited data, current guidelines highlight preoperative TVE [19,20], especially in patients with known or suspected obstructing glottic or supraglottic airway pathologies [20]. Few studies addressed the issue of larynx endoscopy for the prediction of difficult direct laryngoscopy [9,17,18,37,39]; some excluded patients with laryngeal lesions [17,18]. Further, only very few data exist regarding preoperative TVE prior to scheduled videolaryngoscopy [9,45]. We demonstrated that preoperative TVE is beneficial, even if videolaryngoscopy is universally available.

Pharyngolaryngeal lesions are not represented in most traditional bedside airway examination tests [5,6] and thus pose a faithful “blind spot”; our data indicate that TVE might fill this diagnostic gap. TVE-based detection of pharyngolaryngeal lesions might be a game-changer for the decision between awake videolaryngoscopic or bronchoscopic intubation. 

Recently the “TVE score” has been developed. It reuses existing stored TVE recordings to predict difficult airway management and adds incremental diagnostic value to the Mallampati score. Supraglottic, arytenoid, and vestibular fold lesions were concerning. However, in this retrospective study, most patients were handled with direct laryngoscopy [9].

In our current study, dry cough (symptom screening) and vocal cord lesions (TVE finding) were associated with a decreased risk. Importantly, these findings have to be interpreted in the context of the given cohort preselection (only individuals with anticipated difficult airway management) and indicate that individuals with dry cough or isolated vocal cord lesions are at lower risk than those with more serious risk factors (many patients had space-consuming lesions). Moreover, isolated vocal cord lesions are often symptomatic at an early stage and are typically managed with small endotracheal tubes. Vocal cord lesions were very rare findings in our cohort.

Interestingly, while epiglottis lesions were not a relevant risk factor for direct laryngoscopy in the “TVE score” [9], they were a relevant risk factor for videolaryngoscopy in the present study. This might be due to the fact that the epiglottis–blade interaction and the epiglottis mobility are some of the most important factors that define difficult videolaryngoscopy [11,30].

This study has some limitations. Our data represent a single-center experience and caution should be used when extrapolating them to other institutions, since risk stratification, TVE, and videolaryngoscopy techniques might differ. The study has been conducted in a highly specialized center; for the interpretation of study findings, the expertise of the physicians that interpreted the TVE findings has to be taken into account. Our findings can only be extrapolated to patients with suspected difficult airway management scheduled for Macintosh videolaryngoscopy undergoing ENT or OMF surgery. Risk prediction relies on the clinical experience of the responsible anesthetist; here, TVE can only be supportive. The final individualized decision must be made context dependent by a skilled airway operator. Variable selection did not account for multiple observations per individual. Further external validation in lower-risk cohorts could reinforce our findings.

In conclusion, SARI predicts difficult intubation with Macintosh videolaryngoscopes. However, pharyngolaryngeal lesions represent a faithful blind spot in the SARI; TVE has the potential to close this gap in patients with anticipated difficult airways and was superior to a simple symptom screening for pharyngolaryngeal lesions. Vestibular fold and epiglottis lesions, pharyngeal secretion retention and relevant restrictions of the glottis view were concerning. Our data illustrate that TVE is a useful supplement to traditional bedside airway examination tests, allowing advanced risk assessment in order to promote decision making in patients with suspected difficult airway management, even if videolaryngoscopy is universally available at the bedside.

## Figures and Tables

**Figure 1 jcm-12-03433-f001:**
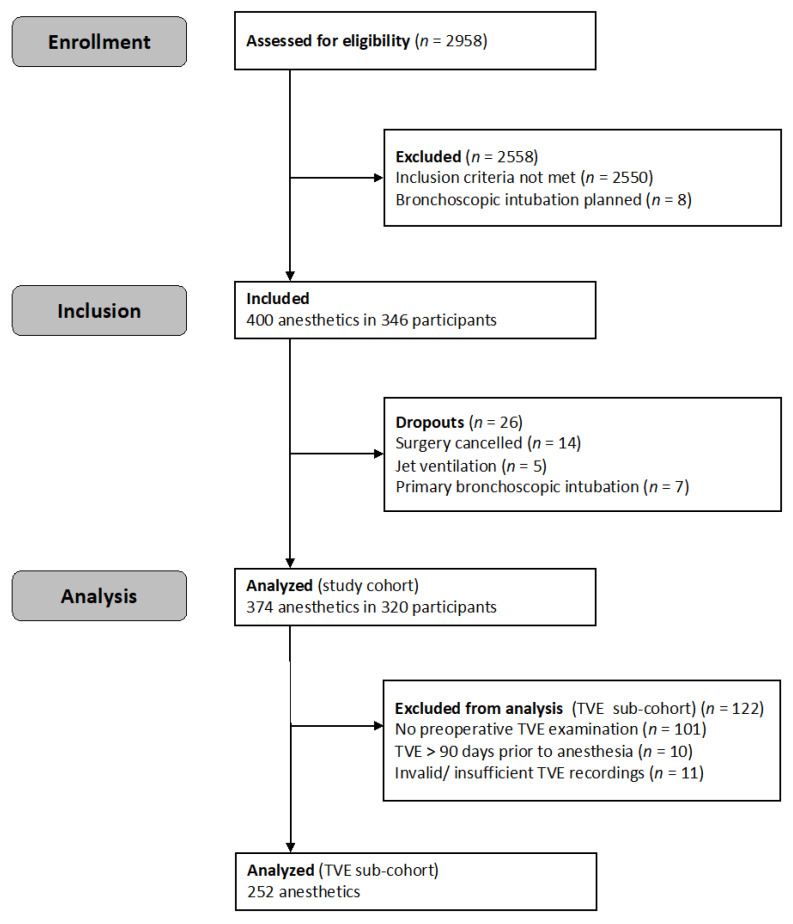
Study flow.

**Figure 2 jcm-12-03433-f002:**
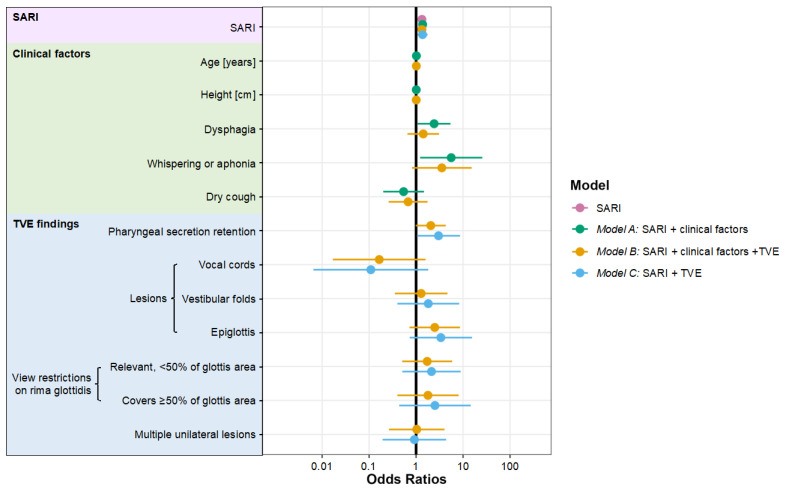
Forest plot of the multivariable mixed effects logistic regression model for the primary outcome measure “difficult videolaryngoscopic intubation alert”. Odds ratios are illustrated as dots on a logarithmic scale (*x*-axis), whiskers indicate 95% confidence intervals. SARI: simplified airway risk index; TVE: transnasal videoendoscopy.

**Table 1 jcm-12-03433-t001:** Patient demographics in the study and TVE sub-cohort.

Characteristics	Study Cohort *n* = 374	TVE Sub-Cohort *n* = 252
Age [years], mean (SD)	61.5 (13.8)	61.8 (13.5)
Sex [male]	69.5% [260/374]	73.4% [185/252]
ASA physical status classification [grade]		
1	5.6% [21/374]	5.6% [14/252]
2	33.7% [126/374]	31.7% [80/252]
3	57.8% [216/374]	58.7% [148/252]
4	2.9% [11/374]	4.0% [10/252]
Previous neck dissection	29.9% [112/374]	26.6% [67/252]
Previous tracheostomy	27.0% [101/374]	27.4% [69/252]
Previous neck radiotherapy	25.4% [95/374]	25.4% [64/252]
Previous awake tracheal intubation	17.1% [64/374]	15.5% [39/252]
Previous mouth floor resection	14.4% [54/374]	11.5% [29/252]
Existing anesthesia alert card	12.8% [48/374]	15.9% [40/252]
Mallampati class		
1	11.2% [42/374]	11.1% [28/252]
2	21.4% [80/374]	23.8% [60/252]
3	31.8% [119/374]	29.4% [74/252]
4	35.6% [133/374]	35.7% [90/252]
Supraglottic tumor	25.1% [94/374]	28.6% [72/252]
Glottic tumor	9.6% [36/374]	11.1% [28/252]
SARI [0–12], median (IQR)	4 (3–6)	4 (2.5–6)
Could not bite upper lip	38.2% [143/374]	33.3% [84/252]
Operation		
Laryngopharyngeal	40.4% [151/374]	44.0% [111/252]
Lower jaw	23.8% [89/374]	24.6% [62/252]
Neck, maxillofacial	20.3% [76/374]	16.7% [42/252]
Ear, nose	9.4% [35/374]	9.9% [25/252]
Dentoalveolar	6.1% [23/374]	4.8% [12/252]
Nasal intubation	30.2% [113/374]	26.6% [67/252]
Rapid sequence intubation	7.8% [29/374]	7.1% [18/252]

ASA: American Society of Anesthesiologists; SARI: simplified airway risk index; TVE: transnasal videoendoscopy; data are presented as mean (standard deviation, SD) or median (interquartile range, IQR); categorical data are presented as percentage values calculated as [frequencies/number of valid data]; ordinal data are presented as median (interquartile range, IQR).

**Table 2 jcm-12-03433-t002:** Primary and secondary outcome measures.

Outcome Measures	Study Cohort *n* = 374	TVE Sub-Cohort *n* = 252
Difficult videolaryngoscopic intubation alert	48.9% [183/374]	55.2% [139/252]
Difficult intubation *	30.5% [114/374]	32.1% [81/252]
Difficult videolaryngoscopy *	19.3% [72/374]	22.2% [56/252]
Transition to a hyperangulated blade	20.3% [76/374]	23.8% [60/252]
Transition to bronchoscopic intubation	1.3% [5/374]	1.2% [3/252]
Laryngoscopy attempts		
1	67.1% [251/374]	63.5% [160/252]
2	24.3% [91/374]	27.4% [69/252]
>2	8.6% [32/374]	9.1% [23/252]
Intubation attempts		
1	69.5% [260/374]	67.9% [171/252]
2	12.8% [48/374]	13.1% [33/252]
>2	17.6% [66/374]	19.0% [48/252]
First pass success †	52.1% [195/374]	50.4% [127/252]
Time to tracheal intubation [s], median (IQR)	86 (42–175)	90 (43–177)
End-tidal pCO_2_ after intubation [mmHg], mean (SD)	36 (8.4)	36 (8.5)
Airway-related adverse events	18.2% [68/374]	18.7% [47/252]
Length of hospital stay (days), median (IQR)	3 (2–7)	3 (2–7)
Deaths in hospital	0.5% [2/374]	0.4% [1/252]

TVE: transnasal videoendoscopy; pCO_2_: partial pressure of CO_2_; data are presented as mean (standard deviation, SD) or median (interquartile range, IQR); categorical data are presented as percentage values calculated as [frequencies/number of valid data]; ordinal data are presented as median (interquartile range, IQR). * As defined previously [19]. † First pass success was defined as only one attempt at laryngoscopy and intubation.

**Table 3 jcm-12-03433-t003:** Co-variables that were selected via LASSO regression analysis (not shrunk to zero) and used for model development in addition to the SARI.

Characteristics	
**SARI**	**(*n* = 374)**
SARI [0–12], median (IQR)	4 (3–6)
**Clinical signs**	**(*n* = 374)**
Age [years], mean (SD)	61.5 (13.8)
Height [cm], mean (SD)	174 (9.6)
Dysphagia	35.3% [132/374]
Weak voice or phonation difficulties	18.7% [70/374]
Whispering or aphonia	8.3% [31/374]
Dry cough	17.9% [67/374]
Productive cough	19.5% [73/374]
Impaired expectoration	13.1% [49/374]
Stridor	1.9% [7/374]
**Transnasal videoendoscopy findings**	**(*n* = 252)**
Pharyngeal secretion retention	47.6% [120/252]
Lesions	
Vocal cords	2.8% [7/252]
Vestibular folds	17.5% [44/252]
Epiglottis	17.5% [44/252]
Multiple unilateral lesions	21.8% [55/252]
View restriction on rima glottidis	
Relevant, covers < 50% of the glottis area	13.9% [35/252]
Covers ≥ 50% of the glottis area	11.5% [29/252]

SARI: simplified airway risk index; data are presented as mean (standard deviation, SD) or median (interquartile range, IQR); categorical data are presented as percentage values calculated as [frequencies/number of valid data]; ordinal data are presented as median (IQR).

**Table 4 jcm-12-03433-t004:** Multivariable mixed effects logistic regression model for the primary outcome measure “difficult videolaryngoscopic intubation alert” (with data from the TVE sub-cohort, *n* = 252).

Characteristics	SARI OR (95% CI)	Model A SARI with Clinical Factors OR (95% CI)	Model B SARI, Clinical Factors and TVE OR (95% CI)	Model C SARI with TVE OR (95% CI)
Likelihood ratio test; compared with SARI, *p*-values	-	*p* = 0.01	*p* < 0.001	*p* < 0.001
Likelihood ratio test; compared with model C, *p*-values	*p* < 0.001	*p* < 0.001	*p* = 0.37	-
**SARI**				
SARI [0–12]	1.33 (1.13 to 1.58)	1.38 (1.15 to 1.65)	1.33 (1.14 to 1.56)	1.37 (1.08 to 1.76)
**Clinical factors**				
Age [years]	-	1.02 (0.99 to 1.04)	1.01 (0.99 to 1.04)	-
Height [cm]	-	1.01 (0.97 to 1.05)	1.01 (0.97 to 1.05)	-
Dysphagia	-	2.42 (1.08 to 5.41)	1.42 (0.66 to 3.09)	-
Whispering or aphonia	-	5.60 (1.22 to 25.74)	3.53 (0.82 to 15.25)	-
Dry cough	-	0.54 (0.20 to 1.47)	0.68 (0.26 to 1.76)	-
**TVE findings**				
Pharyngeal secretion retention	-	-	2.06 (0.98 to 4.29)	3.01 (1.05 to 8.63)
Lesions				
Vocal cords	-	-	0.16 (0.02 to 1.60)	0.11 (0.01 to 1.82)
Vestibular folds	-	-	1.28 (0.35 to 4.66)	1.82 (0.40 to 8.29)
Epiglottis	-	-	2.50 (0.72 to 8.63)	3.37 (0.73 to 15.54)
View restriction on rima glottidis				
Relevant, <50% of glottis area	-	-	1.72 (0.51 to 5.87)	2.13 (0.51 to 8.89)
Covers ≥50% of glottis area	-	-	1.79 (0.40 to 8.04)	2.52 (0.44 to 14.56)
Multiple unilateral lesions	-	-	1.03 (0.26 to 4.03)	0.92 (0.19 to 4.38)
ICC	0.30	0.29	0.21	0.38
AICc	327.1	323.2	316.8	311.0

TVE: transnasal videoendoscopy; SARI: simplified airway risk index; ICC: intra-class correlation; AICc: Akaike information criterion with correction for small sample size; data are presented as odds ratio (OR) with 95% confidence interval (95% CI). A logistic regression analysis for the primary outcome measure (difficult airway alert) was conducted, with the SARI score being the only independent variable. Subsequently, the SARI was used as a component of all multivariable mixed logistic regression models, but also as a comparator.

## Data Availability

Not applicable.

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
