# Peer review of "Nasendoscopy to Predict Difficult Videolaryngoscopy: A Multivariable Model Development Study"

_jcm, 2023, doi:10.3390/jcm12103433_

Round 1
Reviewer 1 Report
The paper by Sasu and Colleagues covers an interesting and actual topic.
English is fluent, and paper design clear and reproducible.
Premises are informative, methods well described, results clear and discussion exhaustive; references are updated and fitting.
Statistics are highly elaborated, and not easy to catch at first glance, but I understand such an approach was required to validate the findings of the study. I would maybe advise to insert a few lines either in the methods or, maybe better, in the discussion section explaining in few simple words the principles adopted for statistical analysis, so to allow a better understanding of statistical tools used for thee non-expert reader. In my opinion understanding the principles will make the reader more “convinced” with results.
I would also emphasize that to prove your concept you selected “almost naturally” difficult airway cases undergoing “difficult” surgeries with shared surgical field, which ads complexity to patients but at same time pushes your evaluation tools to the maximum. It will be interesting to explore whether the same tool has same sensitivity/specificity and usefulness in “presumed easy” airways, which are probably the most frequent ones and those which may benefit of further exploration in “suspicious” cases so to avoid unexpected troubles.
I also agree with declared study limitations, and I would add an extra factor, which is the expertise of physicians working in such a highly specialized clinical center. Some of the results could be “gently” biased by your same expertise (though this does not affect study results).
Nice work.
Author Response
GENERAL COMMENTS: The paper by Sasu and Colleagues covers an interesting and actual topic. English is fluent, and paper design clear and reproducible. Premises are informative, methods well described, results clear and discussion exhaustive; references are updated and fitting.
RESPONSE: Dear reviewer #1, we are delighted and honored that you are interested in our scientific work. Thank you very much for your important comments on our manuscript and great feedback. We revised the manuscript as you recommended.
RECOMMENDATION 1: Statistics are highly elaborated, and not easy to catch at first glance, but I understand such an approach was required to validate the findings of the study. I would maybe advise to insert a few lines either in the methods or, maybe better, in the discussion section explaining in few simple words the principles adopted for statistical analysis, so to allow a better understanding of statistical tools used for thee non-expert reader. In my opinion understanding the principles will make the reader more “convinced” with results.
RESPONSE: Thank you very much for the good advice. We now tried to explain the used statistical methods in some more detail. We amended the method section:
- p4, ln114-115: “The method of Riley et al. [35] was used to calculate the sample size for model development studies (R-package “pmsampsize”).”
- p4, ln135-137: “In a first step, potentially eligible co-variables (predictor variables for model development) were identified by literature review [7,8,12,36-39], previous studies [9,40] and clinical considerations and comprise three categories:”
- p5, ln167-173: “Of note, penalized or regularized regression techniques such as LASSO regression can be used for variable selection to avoid model overfitting and optimism bias. LASSO regression identifies the variables and corresponding coefficients from a set of candidates. This leads to a model with minimized prediction error by imposing a constraint on the model parameters that shrinks the regression coefficients of more irrelevant variables towards zero. Variables with a regression coefficient of zero after shrinkage are excluded [42].”
- p6, ln213-218: “Of note, the AICc is a statistical method to determine the relative quality of a model; hence, AICc is often used to compare different models regarding their goodness of fit in contrast to their simplicity. To achieve this, AICc uses the maximum likelihood estimate and adds a penalty term for the number of independent variables. Thus, it handles the risks of overfitting and underfitting a model resulting in a model with the greatest amount of variation using the fewest possible co-variables [44].”
We further amended the discussion section:
- p10, ln307-309: “We used AICc and LRT to identify a best fitting and performing model for the prediction of difficult airway alerts after videolaryngoscopy based on SARI, clinical factors and TVE findings.”
RECOMMENDATION 2: I would also emphasize that to prove your concept you selected “almost naturally” difficult airway cases undergoing “difficult” surgeries with shared surgical field, which ads complexity to patients but at same time pushes your evaluation tools to the maximum. It will be interesting to explore whether the same tool has same sensitivity/specificity and usefulness in “presumed easy” airways, which are probably the most frequent ones and those which may benefit of further exploration in “suspicious” cases so to avoid unexpected troubles.
RESPONSE: Thank you very much for pointing out this important issue. We agree with you; we believe that the selection of an appropriate study cohort, that reflects the population and clinical situation in which the diagnostic measure is proposed to be used for decision-making in daily clinical practice, is outstandingly important for a diagnostic study. In our opinion the strongest need for preoperative TVE is in ENT/OMF surgery as there is the highest incidence for pharyngolaryngeal lesions and difficult videolaryngoscopic intubations. As TVE is a (minimal) invasive diagnostic measure, it disqualifies as a screening tool in inapparent low risk patients in our opinion. Our data indicate that TVE is a valuable diagnostic tool in suspicious or symptomatic patients with a high pretest probability for pharyngolaryngeal lesions. However, we agree with you that further external validation of our model in lower risk cohorts could reinforce our findings and consider conduction of corresponding follow-up studies soon. We added to the manuscript (p11, ln360): “Further external validation in lower risk cohorts could reinforce our findings.” and (p10, ln312-313) “Our data indicate that TVE is a valuable diagnostic tool in suspicious patients with a high pretest probability for pharyngolaryngeal lesions.”
RECOMMENDATION 3: I also agree with declared study limitations, and I would add an extra factor, which is the expertise of physicians working in such a highly specialized clinical center. Some of the results could be “gently” biased by your same expertise (though this does not affect study results). Nice work.
RESPONSE: Thank you very much for highlighting the importance of expertise and training. Although anesthetists often are very skilled in performing TVE examinations because they use a very similar technique for awake bronchoscopic intubation in daily practice, we agree with you that the interpretation of TVE findings requires some experience. We recommend that the interpretation (but not necessarily the technical skill) should best be done by experienced anesthetists or ENT physicians, if available. We now added to the limitation section (p11, ln352-354): “The study has been conducted in a highly specialized center; for the interpretation of study findings the expertise of the physicians that interpretated the TVE findings has to be taken into account.”
Reviewer 2 Report
The approach is original and methodologically well done. however, the practical implications of TVE to predict a difficult airway will probably not be very significant. every anesthetist knows that existing (anatomic) scores to predict a difficult airway have significant limitations. but usually from the history of the patient (ENT tumors, neck dissection, changing voice etc etc) it is often very clear that a difficult airway is expected. the added value of the proposed score is probably not very high.
Author Response
Dear reviewer #2, we are honored that you are interested in our scientific work and want to hank you very much for your important comments on our manuscript. We revised the manuscript as you recommended.
RECOMMENDATION: The approach is original and methodologically well done. however, the practical implications of TVE to predict a difficult airway will probably not be very significant. every anesthetist knows that existing (anatomic) scores to predict a difficult airway have significant limitations. but usually from the history of the patient (ENT tumors, neck dissection, changing voice etc etc) it is often very clear that a difficult airway is expected. the added value of the proposed score is probably not very high.
RESPONSE: Dear reviewer #2, thank you very much for reviewing our manuscript, for your great feedback and your thoughtful and constructive criticism. We agree with you that it is a serious problem that existing preoperative airway risk prediction tests only have poor diagnostic accuracy and in particular low sensitivity (Detsky ME et al. JAMA. 2019;321:493–503. Roth D et al. Cochrane Database Syst Rev. 2018;5:CD008874.). We were not surprised that a large retrospective study found that more than 90% of difficult tracheal intubations were unexpected (Norskov AK et al. Anaesthesia 2015; 70: 272-81.). We share your opinion that experienced senior anesthetists might be able to anticipate the risk for difficult airway management based on clinical history, known ENT tumors and other pharyngolaryngeal lesions, physical examination etc. in many clinical situations. However, most frequently this knowledge, such as the kind of lesion, location, tumor spread, degree of constriction or view restriction is originated from on previous preoperative TVE examinations. Our impression is that even very skilled anesthetists might profit from this additional information if it comes to decision-making (Rosenblatt W et al. Anesth Analg. 2011;112:602–607). Many patients are clinical inapparent or only present with mild symptoms for pharyngolaryngeal lesions. Especially in these patients TVE examinations might be particularly useful to identify those individuals at risk. Our data suggest that TVE findings correlate better with the difficulty of videolaryngoscopic intubation compared with clinical signs, symptoms or functional impairments. We now added to the limitation section (p11, ln356-359): “Risk prediction relies on the clinical experience of the responsible anesthetist; here, TVE can only be supportive. The final individualized decision must be made context-dependent by a skilled airway operator.”